# The Impact of Training in Transgender Care on Healthcare Providers Competence and Confidence: A Cross-Sectional Survey

**DOI:** 10.3390/healthcare9080967

**Published:** 2021-07-30

**Authors:** Aisa Burgwal, Natia Gvianishvili, Vierge Hård, Julia Kata, Isidro García Nieto, Cal Orre, Adam Smiley, Jelena Vidić, Joz Motmans

**Affiliations:** 1Transgender Infopunt, Ghent University Hospital, 9000 Ghent, Belgium; aisa.burgwal@uzgent.be; 2Women’s Initiative Supportive Group (WISG), Tbilisi 0105, Georgia; natiagvianishvili@gmail.com; 3Swedish Federation for Lesbian, Gay, Bisexual, Transgender, Queer and Intersex Rights (RFSL), 116 41 Stockholm, Sweden; vierge@rfslungdom.se (V.H.); Cal.orre@rfsl.se (C.O.); 4Trans-Fuzja, 00-666 Warschau, Poland; julia.kata@transfuzja.org; 5Daniela Fundación, 28002 Madrid, Spain; isidrogarcianieto@gmail.com; 6Transgender Europe (TGEU), 12435 Berlin, Germany; adam.smiley@gmx.de; 7Geten, 11000 Belgrade, Serbia; vidic.jelena@gmail.com

**Keywords:** transgender people, healthcare providers, training, confidence, competence

## Abstract

All studies to date demonstrate a lack of access to care for transgender people. A few educational efforts in providing care to transgender people have been successful. However, one challenge in administering training is that there is almost no research on the need of healthcare providers (HCP) to acquire knowledge, as well as on the effect of training on their level of competence and confidence in working with transgender people. Results from an online survey of a convenience sample of HCP across four different European countries (*N* = 810) showed that 52.7% reported experiences with some form of training on transgender people. The mean confidence level for all HCP (with or without training) in working with transgender people was 2.63, with a significant effect of training on confidence. 92.4% of HCP believed that training would raise their competence, and this belief was significantly higher among HCP with training experience, HCP working in Serbia and Sweden and/or among those HCP who belong to a sexual minority group. General practitioners had the lowest confidence levels of all professions involved. The study provided strong support for the use of training in improving healthcare conditions for transgender people, not only to raise awareness among HCP, but also to increase knowledge, competence and confidence levels of HCP in working with transgender people.

## 1. Introduction

Transgender people experience a variety of challenges in gaining access to healthcare, and it is clear that they experience profound health disparities compared with the general population [1,2,3,4,5,6,7,8,9,10,11]. With the term trans(gender) people we refer to those people who identify with a gender opposite to their sex assigned at birth or identify outside the gender binary of male or female (non-binary and genderqueer individuals). Lack of knowledge and skills on the part of HCP about how to deliver competent care to transgender people has been cited as contributing to these health disparities [2,6,12,13]. Training may evolve as a strategy to reduce such barriers to service delivery [2,14]. In addition, recent publications call for improving the quality of care for transgender people [15,16,17].

All of the transgender health needs assessments that have been conducted to date demonstrate a lack of access to care [3,6,15,18,19,20]. A lack of insurance for trans-specific treatments, a restrictive hormone policy, insufficient institutional support, a lack of knowledge and insensitivity or hostility on the part of HCP are frequently mentioned barriers to care [13,21,22]. For example, the needs assessment of Sperber et al. [20] found that HCP, when encountering a transgender person, often refer to transgender issues or to the person’s gender identity when treating unrelated conditions. This behavior indicates that insensitivity of healthcare providers towards transgender people still occurs. All studies on transgender health needs assessments recommend the development of educational programs or training programs to increase awareness, knowledge and skills in transgender care [2,7,8,23].

Studies have called for increasing provider training as a key strategy in increasing quality of care to transgender people [15,16,24,25,26]. However, one of the main challenges in administering training is that there is almost no research on the need of HCP to acquire knowledge or on how to integrate this into practice [27,28,29,30]. For example, Morrison, et al. [31] have conducted a study with 154 program directors of urologists and plastic surgeons in training. The study showed that more than seven out of ten program directors believed that it was important to have transgender-focused education and 72% emphasized the importance of training opportunities. However, the median amount of training in their programs was approximately one didactic hour and two clinical hours dedicated to transgender content within their medical program. Other studies report similar time dedicated to transgender-related content [32,33].

Two types of training in regard to of transgender (including non-binary and genderqueer) individuals are found within the literature. The first consists of training focusing on the broader group of lesbian, gay, bisexual and transgender (LGBT) individuals, which are not generally integrated into the curriculum for health professionals. This training usually provides more general information, with a clear distinction between sexual orientation and gender identity. The second are types of training which educate providers about how to interact with specifically transgender individuals. Although this training is often more specific and pays more attention to the specific needs of transgender people, it is even less integrated into the curriculum for HCP [30]. This lack of education and prioritization also constitutes an implicit message that expertise in transgender health is optional for HCP and that only those who need to acquire this knowledge will actively seek it out. However, based on existing standards of care such as the Standards of Care 7 (SOC7) from the World Professional Association for Transgender Health (WPATH) [34,35,36], all therapists and HCP can help individuals with exploring and affirming their gender identity, exploring different options for expression of that identity, and making informed decisions about medical treatment options. Basic guidelines tailored at giving clinicians an up-to-date overview of clinical consensus statements on transgender healthcare are therefore necessary (see, for example, [36]).

A few educational efforts to increase competence, knowledge and skills in working with transgender people (or the broader LGBT population) have been successful [24,25,37,38,39,40,41,42,43]. However, each study examines a different type of training, and overall very little follow-up research is available. For example, the study of Hanssmann et al. [24] assessed the effectiveness of three types of training administered by a nonprofit health education and outreach organization serving LGBT people. Quantitative data indicated a statistically significant gain in self-assessed knowledge, and qualitative data confirmed this finding. The study of White Hughto et al. [40] delivered a group-based intervention based on the Theory of Planned Behavior [44] and the Information, Motivation, and Behavioral Skills models [45]. This training targeted correctional HCP cultural competence (e.g., by covering terminology and transgender discrimination) and clinical competence (e.g., by covering the durability of gender identity and hormone treatment regimens to increase knowledge and willingness to provide hormones). Providers’ willingness to provide gender-affirming care improved immediately post-intervention, as well as transgender cultural competence, medical gender affirmation knowledge, self-efficacy in initiating hormones, and subjective norms related to transgender care. However, no information is available on the duration of such a training effect.

When examining existing training programs, we notice some overlap in content and approach, providing several recurring components such as (1) terminology, information, and background about the transgender community; (2) the distinction between sexual orientation and gender identity; and (3) clinical information about unique healthcare needs, as well as strategies on how to conduct clinical interactions in a respectful, affirming manner (see e.g., [25,38,39,41,43]). For example, Lelutiu-Weinberger and Pachankis [25] conducted a 2-day training for mental health professionals in Bucharest, Romania. They based their training on already existing training programs in the U.S., adapted to the Romanian context. The training addressed all three components mentioned above, but put more emphasis on the existence and impact of structural stigma towards transgender people, given the Romanian context of high stigma towards LGBT people and relatively low knowledge regarding sexual and gender diversity.

Sources of contact are commonly offered as well, such as referral networks and support groups within the transgender communities. The presence of a trainer who identifies as transgender can have both a positive and negative effect on HCP. On the one hand, including transgender people in training may benefit providers since they can correct false stereotypes about what a transgender person look like. On the other hand, it can also lead to an exclusionary tendency towards people that do not match with the appearance of the transgender person used in the training [24]. Hanssmann et al. [24] also highlighted that training must specifically articulate that transgender groups and communities include people of color and that, because of racism, these individuals may experience additional barriers to care. This suggests that an intersectional lens is required within transgender training programs, one that pays attention to the interconnected nature of social categorizations such as race, gender, and class [46,47].

Despite the fact that there are indications that training programs do have a positive effect, little is known about whether HCP at large are interested in education of this kind, especially in Europe. U.S. research evaluating the effect of training shows that HCP are interested in training in transgender care. However, this has only been studied among HCP who already participated in training [48]. For example, a 2020 U.S. cross-sectional, electronic survey research, with 153 providers practicing gender-affirming care, was the first to identify type of training experiences and recommendations for best practices in training future providers. This study found that many providers independently acquired skills by reading the literature and existing guidelines (57.3%), but the two most commonly available training opportunities were professional conferences (57.3%) and mentorship (41.3%). Respondents would most recommend training others in their field through structured clinical experience (e.g., rotation or longitudinal exposure during training), rather than additional didactic training. Of respondents, 28% also reported learning about gender-affirming care because it was an expectation in their work environment, with no other motivating factors for seeking training in gender-affirming care [49]. The U.S., cross-sectional survey of Greene et al. [48] among (dental) medical students and nursing students showed that the students reported interest in receiving formal LGBT health education. However, fewer than 50% agreed that their formal training had prepared them sufficiently.

This study aims to disentangle the needs of HCP (ranging from surgeons to administrative staff within care) towards training in transgender care. Past experiences with training, and the desired formats of future training programs are examined, with a specific focus on self-reported level of confidence in working with transgender people, taking into account other variables that might also be of importance in explaining the variance in confidence. Based on the literature, it is hypothesized that training experience will in itself lead to significantly higher confidence levels in working with transgender people. Because the literature is not clear about the overall need of HCP regarding training in transgender care, no predictions will be made here.

## 2. Materials and Methods

### 2.1. Study Population and Procedure

In 2016, an online anonymous survey was sent to HCP (ranging from surgeons to administrative workers) working in one of five European countries (Georgia, Poland, Serbia, Spain and Sweden). Based on previous country-based community-driven research as well as a literature review, an English questionnaire was co-created by the research group consisting of Transgender Europe, and transgender organizations from Georgia, Poland, Serbia, Spain and Sweden, together with the final author, a social scientist. The five countries were chosen based on different criteria: whether or not the country was a member of the EU, the geographical spread across Europe, a well-developed transgender healthcare system or not, the socio-legal position of transgender people in society (for instance regarding legal gender recognition) and the amount of experience in community driven research projects. The questionnaire consisted of open and closed questions, and not all questions were obligatory, resulting in different response rates per question. Existing and validated measurement tools were used for selected topics of interest where possible, such as the International Standard Classification of Education (ISCED-2011; [50]) and the Beliefs about Gender Scale [51]. The final questionnaire was translated into Georgian, Polish, Serbian, Spanish, and Swedish by native speakers, and tested by volunteers in the respective countries. The questionnaire proved to have sufficient face validity and content validity. All scales were also tested on internal consistency and proved to have good reliability and construct validity (0.7 ≤ α ≤ 0.9). The six surveys were hosted on an online survey platform SurveyMonkey, and were accessible between November 2016 and February 2017. HCP currently working in Georgia, Poland, Serbia, Spain, or Sweden were invited to complete the anonymous survey. Snowball sampling was used to reach out to respondents, using social media (open and closed Facebook groups specifically aimed at certain groups of HCP) and the networks in transgender care of the organizations involved in the study.

### 2.2. Main Outcome Measures

Participants were asked a number of demographic questions. Age was recoded, first asking for birth year. Sex assigned at birth (SAAB) was measured with one question asking respondents for their sex assigned at birth, with the explanation that we meant their sex on their initial birth certificate (Female/Male) (as no other legal options existed in the countries under study). Gender identity was measured by asking all respondents how they would describe their gender identity at the current moment. A closed list of possibilities for self-identification was presented, from which they were asked to select only one option that fits them best: Female, Male, Transfeminine/Transgender Woman/Male-to-female (MTF), Transmasculine/Transgender Man/Female-to-male (FTM), Nonbinary/Genderqueer/Gender nonconforming and Other (please specify). If their SAAB was male, and their gender identity was Female, the respondent was recoded into Transgender Woman. If their SAAB was female, and their gender identity was Male, the respondent was recoded into Transgender Man. Transgender women, Transgender men, and Nonbinary/Genderqueer/Gender non-conforming respondents were recoded into transgender HCP. The open answers of those respondents who indicated having an “other” gender identity were screened by the research group, and recoded into one of the gender identity groups (transgender versus cisgender) or removed from the dataset. HCP registered their country in which they work, with the options “Georgia”, “Poland”, “Serbia”, “Spain”, “Sweden”, “Another country”. Belonging to different minority groups was measured with a question where the respondents had to indicate whether they felt they belonged to a minority group in their country of residence (“No, I don’t belong to this group”, “Yes, but it is not important at all to me”, “Yes, but it’s only slightly important to me”, and “Yes, and it’s very important to me”). The listed minority groups were: ethnic minority, religious minority, sexual minority (gay, lesbian, bisexual, asexual, etc.) and minority due to ability status. For each minority group respondents were recoded into a binary variable indicating whether or not they felt they belonged to that specific minority group (1 = No, 2 = Yes). The difference in the evaluation of the importance of belonging to a minority group was not used due to small sample sizes (*n* < 20). Finally, HCP were asked to indicate their specific type of profession from a predefined list (Yes/No).

Experience with training about transgender people, transsexualism, or gender dysphoria (Yes/No) was assessed. The terms transsexualism (ICD-10, [52]) and gender dysphoria (DSM-5 [53]) were used because they were regularly utilized in the countries under study at the time of the survey. HCP with training experience were asked how the training was provided, who provided the training, and in what format the training was received. For every question, multiple responses were possible. Four options were possible as to how the training was provided: “As part of my mandatory formal education program”, “As part of my mandatory professional development”, “Voluntarily on my own initiative”, “Other (please specify)”. Responses to who provided the training were organized in six options: “A professional healthcare provider from outside the university”, “A trans- or LGBTI organization”, “An instructor through the university”, “City/county/government or administration”, “An employer”, “Other (please specify)”. At last, the format of the training was examined through five options: “As a topic in a course book”, “As a subject of a lecture or a topic within a course I attended”, “As a topic of a workshop, seminar, or conference”, “Online or web-based course”, “Other (please specify)”.

Needs in training was assessed by asking all respondents, regardless of training experiences, what type of training would be helpful to increase levels of competence, by whom this training should be provided, and in what format. For every question, multiple responses were possible. For ‘type of training’ and for ‘whom they would like to receive training from’, the same answer options were used as described above. The question about the format of the training was described slightly differently, with six options to choose from: “In the form of course books”, “In the form of testimonies by transgender people”, “Online or web-based course”, “As a course organized by a healthcare provider specialized in trans-specific healthcare”, “As a workshop or seminar organized by a transgender organization”, “Other (please specify)”.

Knowledge in regards to referral networks and existing guidelines, as well as protocols at their workplace were measured with multiple questions. First, a question was asked whether respondents knew where to refer transgender clients to when they wish to access a type of trans-specific healthcare or a transgender support group (1 = Yes, 2 = No, 3 = No, but I know where I could get the information). Second, knowledge about the existence of a protocol/guidelines for transgender-specific care in their respective country was asked for (1 = Yes, 2 = No, 3 = I don’t know). Additionally, it was asked which of the following six guidelines were used by the HCP: the Standards of Care 5, 6 or 7 from WPATH, ICD-10, DSM-IV or DSM 5. Other options were “None of the above” and “Other (please specify)”. Multiple answers were possible here. Guidelines at their workplace for transgender inclusive health services were assessed first by asking about guidelines in their current work for pronoun use of healthcare users. Four answer options were listed: “Yes, there are existing guidelines”, “Guidelines are currently being developed”, “No, there are no guidelines”, and “I don’t know”. Second, four measures were provided and respondents were asked if these measures were taken in the workplace of the healthcare provider (1 = Yes, 2 = No, 3 = I don’t know): gender-neutral toilets, privacy at the reception desk, alternatives to calling the legal name in the waiting room, and alternatives to listing legal names in the digital database.

Confidence level in working with transgender people accessing healthcare was measured by self-reported level of confidence on a five point Likert scale, ranging from 1 “Very high” to 5 “Very low”. HCP were also asked if in their opinion, their level of competence in working with transgender people would increase with training (Yes/No).

### 2.3. Statistical Analysis

All statistical analyses were conducted using SPSS for Windows, v26.0 [54], with statistical significance determined at *p* < 0.05. Descriptive statistics were obtained for all variables (proportions, frequencies, means, standard deviations). Categorical variables (such as SAAB, country, belonging to a minority group, and profession) were dummy coded for analysis. Demographic differences between HCP were analyzed using *t*-tests (continuous variables) and Chi-square analyses (categorical variables). To test the association between past training and confidence in working with transgender people, analysis of covariance (ANCOVA) was applied, attempting to obtain a model, using a stepwise regression strategy. A backward stepwise selection procedure was applied to a model with all demographic background variables (age, SAAB, profession, country of residence, belonging to an ethnic, religious, sexual and/or disability minority) and past training experience. Stepwise selection indicates variables with a statistically significant effect, simultaneously adjusting for the other variables in the regression model [55]. To avoid the problems associated with automatic variable selection procedures with the stepwise approach [56], a backward stepwise selection was applied manually. Variance inflation factors (VIF) were calculated for each of the variables included in the models, and were dealt with when VIFs were too high (VIF > 4) [57].

## 3. Results

### 3.1. Response and Demographic Background

The data-cleaning process excluded respondents who did not give their consent (*n* = 3), who were not working in one of the five countries under study (Georgia, Poland, Serbia, Spain, or Sweden, *n* = 23), and those who indicated having a transgender identity (transgender man, transgender woman or non-binary gender, *n* = 52). Respondents from Georgia were not used within this article since the response rate was too low (*n* = 16). Only cisgender HCP were maintained within analysis, since it is assumed that transgender HCP (*n* = 52) are more informed about transgender-specific healthcare. After data cleaning, the data contained answers from *N* = 810 respondents. Demographics are given in Table 1, including number of participants endorsing the category with the valid percentage reported. All professions were grouped into four categories, specifically “general practitioners”, “nurses”, “psycho-social care providers”, and “other medical specialists” (see Table 1).

### 3.2. Training Experiences and Training Needs

Notably, almost half of all participants (52.7%, *n* = 427) indicated ever having had some sort of training about transgender people, transsexualism, or gender dysphoria, with a significant difference in training experiences between professions (*X*^2^(3) = 31.73, *p* < 0.001). Almost seven out of ten psycho-social care providers indicated having had training experience (69.2%) versus only 41.9% of general practitioners, 47.2% of nurses and 50.4% of respondents from another medical profession. The participants who had training experience indicated that their training background was mostly the result of voluntarily seeking information (59.3%, *n* = 249), most often provided by a transgender or LGBTI organization (48.2%, *n* = 201), and/or as a topic of a workshop, seminar, or conference (60.1%, *n* = 250) (see Table 2).

Most participants indicated that they would prefer the training to be part of their mandatory professional development (63.0%, *n* = 455), provided by a trans- or LGBTI organization (80.6%, *n* = 580), and/or as a course organized by a healthcare provider specialized in trans-specific healthcare (78.9%, *n* = 569) (see Table 3).

### 3.3. Knowledge of Referral Networks and Guidelines

Current knowledge of HCP was mapped in regards to referral networks, and existing protocols. Additionally, we inquired about current guidelines for transgender inclusive health services at the workplace. A summary of the results are presented in Table 4.

More than one in five respondents (21.7%, *n* = 166) indicated they did not know where to refer their transgender client to, when they wanted to access a type of transgender-specific healthcare which is not offered by the healthcare provider themselves. Another 32.6% (*n* = 250) did not know where to refer to, but did know where to get this information. The same applied for referral networks to contact a transgender support group; 24.0% (*n* = 183) did not know where to refer to, and 31.1% (*n* = 237) did know where to get the information.

Furthermore, 41.3% (*n* = 315) did not know if there was a protocol in place for transgender-specific care (on a national or regional level). Almost half of the respondents (45.6%, *n* = 334) did not use any guideline or manual for transgender people accessing healthcare. Noteworthy is that a high amount of HCP used outdated manuals such as the DSM IV (14.1%, *n* = 103), the Standards of Care 5 from the WPATH (1.6%, *n* = 12) or the Standards of Care 6 from the WPATH (1.9%, *n* = 14).

Regarding current transgender inclusive guidelines in their workplace, more than half (54.3%, *n* = 393) indicated there were no guidelines in place at work for name and pronoun use, and 18.8% (*n* = 136) did not know if there were any guidelines. More than half of all respondents (65.0%, *n* = 460) indicated that there was no privacy at the reception desk and 54.0% (*n* = 383) indicated that there were no alternatives in place to listing legal names in their digital databases. Of the total, 51.2% (*n* = 366) had no gender-neutral toilets for clients, and another 45.1% (*n* = 321) had no alternatives to calling out the legal name in the waiting room.

### 3.4. Level of Confidence

Mean level of confidence in working with transgender people accessing healthcare was 2.63 (SD = 1.00), ranging between an average and high level of confidence. A significant difference was found among the participants who had received training and those who had not (*t*(618.89) = −8.12, *p* < 0.001). The mean confidence level was significantly higher for the group with previous training experience compared to the group with no training, suggesting that training increases confidence levels in working with transgender people accessing healthcare. When asked if HCP believed that level of competence in working with transgender people accessing healthcare would increase by means of training, 92.4% (*n* = 733) believed this to be true.

A backward regression analysis method was applied to test the hypothesis of a significant difference in confidence between HCP with and without training, when controlling for different socio-economic variables and professional groups. Firstly, two-way ANOVAs were performed, in order to see any significant interaction effects between training and one of the demographic variables. Then, training, as well as country of residence, sex assigned at birth, age, profession, and belonging to one/more minority groups (ethnic, sexual, religious and/or disability) were taken together in one model. Two significant interaction effects based on two-way ANOVAs, between on the one hand training and on the other hand country of residence and belonging to a sexual minority, were also included in the first model.

Using the stepwise selection procedure, all significant effects were taken together in one model (see Table 5). The first model enters all candidate variables, the two significant interaction effects included. Within each step, the variable with the highest *p*-value was excluded until only variables with a *p*-value < 0.05 remained. The eighth model is chosen as the final model. All variables in model six also have *p*-values less than 0.05. It appears to be a good model fit, but Variance Inflation Factors (VIFs) were to high (> 10) between training, country and the interaction term included. In order to determine the precise effect of each predictor, the interaction term was removed from the model.

HCP with no training continued to show significantly lower confidence levels than HCP with training experience (F (1, 738) = 59.73, *p* < 0.001), thus a significant main effect for training remained.

Furthermore, there was a significant main effect for country (F (3,738) = 3.88, *p* = 0.009) and profession (F (3, 738) = 10.59, *p* < 0.001). Post hoc comparisons using the Bonferroni correction indicated that confidence was on average significantly lower for respondents working in Spain (M = 3.00, SD = 1.21) in comparison to respondents working in Serbia (M = 2.46, SD = 1.05) or Sweden (M = 2.48, SD = 0.84). General practitioners (M = 3.23, SD = 1.03) had significantly lower confidence levels than nurses (M = 2.56, SD = 0.88), psycho-social care providers (M = 2.45, SD = 0.83), and other medical specialists (M = 2.51, SD = 1.06). HCP belonging to a sexual minority group (gay, lesbian, bisexual, etc.) also reported higher confidence levels in working with transgender people accessing healthcare (F (1, 738) = 12.64, *p* < 0.001).

## 4. Discussion

Training in transgender care is an important component in the process of bringing to light how providers can increase quality of care to transgender individuals. Such training can familiarize HCP with the barriers these groups encounter in gaining access to care (lack of knowledge, inappropriate curiosity, disrespect, discrimination, etc.) [6,9,10,11]. They can also provide a context in which providers can improve their skills at delivering care to transgender individuals, by providing networks for referral and information about support groups. The current study demonstrates a lack of knowledge about referral networks and support groups. If professional guidelines are used, they are often outdated.

The present study depicts an initial scoping of training experiences of HCP and differences between HCP with and without training, taking into account the impact of several socio-economic positions (country, belonging to a minority group, SAAB, age, profession). In our sample, 52.7% indicated having had training experiences. The hypothesis suggesting a significant difference between HCP with and without training on confidence, when controlling for different socio-economic positions, was confirmed. Indeed, HCP with training experience reported significantly higher confidence in working with transgender people accessing healthcare in comparison to HCP without training experience. Additional influences of country, profession and belonging to a sexual minority group were found. Confidence was lower for respondents working in Spain in comparison to respondents working in Serbia or Sweden. General practitioners had lower confidence levels than nurses, psycho-social care providers, and other medical specialists. HCP belonging to a sexual minority group (gay, lesbian, bisexual, etc.) also reported higher confidence levels in working with transgender people accessing healthcare. 

HCP also indicated favoring training as part of their mandatory professional development, provided by a transgender-specific or LGBTI organization, and/or as a course organized by a healthcare provider specializing in transgender-specific healthcare. Such training exists, for example the WPATH’s Global Education Initiative (GEI) offers certified training courses to HCP in the context and principles of the WPATH Standards of Care (SOC7, [34]) and on their implementation into clinical practice. In line with the study of Stryker et al. [49], most HCP indicated to have gained information voluntarily on their own initiative (59.3%) and/or at conferences (60.1%). Stryker et al. [49] proposed structural clinical experiences as the preferred means of training and not didactic training. Didactic training in the format of course books was not preferred within this study. Testimonies of transgender people were preferred by 63.4% of HCP. As Hanssmann et al. [24] pointed out, this may benefit HCP since it can correct false stereotypes about what a transgender person look like, yet it can also lead to an exclusionary tendency towards people who do not match the appearance of the transgender person used in the training. Visual materials such as pictures can also be used within training (see, for example, the Gender Spectrum Collection [58]).

The fact that healthcare providers indicated that they mainly received training on a voluntary basis, but preferred training within their mandatory professional development, advocates for the inclusion of such trainings in transgender care within mandatory educational curricula.

Currently, there is no broadly accepted curriculum included in educational courses within Europe with regard to transgender health issues. In the U.S. some efforts have been made to educate students about transgender care (see, for example, [48]).The results from this study should be used when developing such training programs, taking into account previously conducted training that proved to be successful [24,25,37,38,39,40,41,42,43]. Future research should focus on the development, as well as long-term follow-up, evaluation and adaptation, of a training program, taking into account the intersections of, among others, sexual orientation, race and class. Also, in order for training to lead to sustainable improvements in quality of care, it should be accompanied by organizational or agency-wide change and support.

This study found that more than half of the respondents indicated that no measures (e.g., no privacy at the reception desk, no alternatives in place to listing legal names in the database) were taken to provide a transgender affirming climate. This aligns with the fact that many transgender individuals delay access to care (see, e.g., Burgwal and Motmans [59]). The survey study of Burgwal and Motmans [59] indicated that transgender people give many reasons for not accessing general and trans-specific healthcare services (e.g., afraid of being treated badly, the complexity of the bureaucracy, availability of healthcare services). Thus, concrete policies with regard to quality assurance must be applied, not just to ensuring that HCP have the tools and skills to deliver supportive and competent care to transgender individuals, but to focus future research on how to induce change on different levels.

A limitation is that this study may have produced data that is skewed due to the sampling strategy. The survey was distributed through relevant online networks, which could explain why the majority of respondents were younger than 40, AFAB, and lived in a city, the suburbs, or outskirts of a city. Research notes that AFAB people participate more often in online surveys that are characterized by communication and information exchange, while AMAB more often participate in online surveys that are characterized by the search for information [60,61]. Responding to an online survey is a process of online information exchange, so it is reasonable that a higher response rate was observed among AFAB participants than AMAB participants. HCP working in rural areas may not have been reached. Another limitation within the entire study concerned the total respondents per country, which differed greatly and led to the decision to exclude data from Georgia as a control variable. The significant difference between countries indicates that future international research is needed to take a country’s climate into account.

## 5. Conclusions

This study provided strong support for the use of training in improving healthcare conditions for transgender people, not only to raise confidence levels of HCP in working with transgender people, but to improve transgender-specific healthcare conditions in general. Training experiences proved to be significantly associated with higher confidence levels of HCP, with an additional influence of country of residence, profession, and belonging to a sexual minority group. Most HCP indicated the desire for training within their mandatory professional development, which demonstrates the needs of HCP regarding training in transgender care. Quantitative data analysis is promising in terms of showing an increase in knowledge and competence in healthcare delivery to transgender individuals as a result of training, but more long-time follow-up research is needed in this area to provide a training manual applicable to HCP.

## Figures and Tables

**Table 1 healthcare-09-00967-t001:** Demographic variables (*N*(%) for categorical variables, M(SD) for continuous variables).

Demographics	*N*	Valid %
**Country of work**		
Poland	87	10.7
Serbia	55	6.8
Spain	223	27.5
Sweden	445	54.9
**Sex assigned at birth**		
Female	648	80
Male	162	20
**Belong to minority groups (Yes)**		
Ethnic	48	5.9
Religious	61	7.5
Sexual	173	21.4
Disability	33	4.1
**Current Profession**		
**General practitioner**	148	18.3
**Nurse**	195	24.1
**Psycho-social care providers**		
Psychologist	86	10.6
Psychotherapist	36	4.4
Sexologist	28	3.5
Counsellor	51	6.3
**Other medical specialist**		
Pregnancy and post-natal care	40	4.9
Pediatrician	38	4.7
Social worker	26	3.2
Geriatric care	23	2.8
Other medical specialist	21	2.6
Administrative or clerical staff	15	1.9
Psychiatrist	11	1.4
STI testing personnel	12	1.5
Endocrinologist	8	1.0
Gynecologist	7	0.9
Surgeon	5	0.6
Reproductive health specialist	3	0.4
Plastic surgeon	3	0.4
Dentist	2	0.2
Physical therapist	2	0.2
Urologist	1	0.1
Other	49	6.0
**Age**	**M (SD)**	**Min–Max**
	42,23 (11,84)	20–77

M = Mean, SD = Standard deviation, Min = Minimum, Max = Maximum.

**Table 2 healthcare-09-00967-t002:** Training experiences *N* (%).

	*N*	% of Cases
**Training**		
Yes	427	52.7
**Type of training**		
As part of my mandatory formal education program	87	20.7
As part of my mandatory professional development	113	26.9
Voluntarily on my own initiative	249	59.3
Other (please specify)	77	18.3
**Provided by**		
A professional healthcare provider from outside the university	158	37.9
A trans- or LGBTI organisation	201	48.2
An instructor through the university	124	29.7
City/county/government or administration	72	17.3
An employer	27	6.5
Other (please specify)	66	15.8
**Format**		
As a topic in a course book	83	20
As a subject of a lecture or a topic within a course I attended	220	52.9
As a topic of a workshop, seminar, or conference	250	60.1
Online or web-based course	45	10.8
Other (please specify)	50	12

**Table 3 healthcare-09-00967-t003:** Training needs *N* (%).

	*N*	% of Cases
**Type of training**		
As part of my mandatory formal education program	383	53
As part of my mandatory professional development	455	63
Non-compulsory training opportunities	394	54.6
Other (please specify)	39	5.4
**Provided by**		
A professional healthcare provider from outside the university	418	58.1
A trans- or LGBTI organisation	580	80.6
An instructor through the university	296	41.1
City/county/government or administration	134	18.6
An employer	59	8.2
Other (please specify)	47	6.5
**Format**		
In the form of course books	167	23.2
In the form of testimonies by transgender people	457	63.4
Online or web-based course	178	24.7
As a course organized by a healthcare provider specialized in trans-specific healthcare	569	78.9
As a workshop or seminar organized by a trans- or LGBTI organization	472	65.5
Other (please specify)	17	2.4

**Table 4 healthcare-09-00967-t004:** Knowledge of HCP *N* (%).

Knowledge	*N*	Valid %
**Referral trans-specific healthcare**		
Yes	350	45.7
No	166	21.7
No, but I know where to get the information	250	32.6
**Referral transgender support group**		
Yes	342	44.9
No	183	24
No, but I know where to get the information	237	31.1
**Existence protocol for trans-specific care**		
Yes	334	43.8
No	114	14.9
I don’t know	315	41.3
**Usage of guidelines**		
Standards of Care 5 from WPATH	12	1.6
Standards of Care 6 from WPATH	14	1.9
Standards of Care 7 from WPATH	53	7.2
ICD 10	318	43.4
DSM 4	103	14.1
DSM 5	220	30.1
None of the above	334	45.6
Other (please specify)	44	6
**Existence guidelines for pronoun use**		
Yes	159	22
Currently being developed	36	5
No	393	54.3
I don’t know	136	18.8
**Measures taken**		
Gender-neutral toilets	291	40.7
Privacy at the reception desk	152	21.5
Alternatives to calling the legal name in the waiting room	203	28.6
Alternatives to listing legal names in the database	86	12.1

**Table 5 healthcare-09-00967-t005:** Regression analysis for level of confidence (standardized regression coefficients).

Variables	Models							
	1	2	3	4	5	6	7	8
Training (yes)	−0.20 ***	−0.20 ***	−0.20 ***	−0.36 ***	−0.36 ***	−0.36 ***	−0.52 ***	−0.53 ***
Country (Poland)	0.25	0.24	0.22	0.21	0.21	0.22	0.12	0.15
Country (Serbia)	−0.11	−0.14	−0.13	−0.14	−0.12	−0.12	−0.24	−0.24
Country (Spain)	0.48 ***	0.47 ***	0.47 ***	0.49 ***	0.49 ***	0.49 ***	0.22 *	0.21 *
AFAB	0.08	/	/	/	/	/	/	/
Profession (Other medical specialist)	−0.56 ***	−0.56 ***	−0.56 ***	−0.57 ***	−0.56 ***	−0.56 ***	−0.60 ***	−0.59 ***
Profession (psycho−social care provider)	−0.50 ***	−0.50 ***	−0.50 ***	−0.50 ***	−0.50 ***	−0.49 ***	−0.53 ***	−0.51 ***
Profession (nurse)	−0.50 ***	−0.50 ***	−0.49 ***	−0.49 ***	−0.48 ***	−0.48 ***	−0.53 ***	−0.51 ***
Ethnic minority (no)	0.16	0.16	0.19	0.19	/	/	/	/
Religious minority (no)	0.14	0.14	/	/	/	/	/	/
Sexual minority (no)	0.40 ***	0.42 ***	0.42 ***	0.31 ***	0.32 ***	0.33 ***	0.33 ***	0.30 ***
Minority due to ability status (no)	0.34	0.34	0.35 *	0.33	0.34	/	/	/
Age	−0.01 *	−0.01 *	−0.01 *	−0.01 *	−0.01 *	−0.01 *	−0.01	/
Training (yes) * Country (Poland)	−0.23	−0.23	−0.23	−0.19	−0.18	−0.17	/	/
Training (yes) * Country (Serbia)	−0.17	−0.13	−0.16	−0.18	−0.16	−0.16	/	/
Training (yes) * Country (Spain)	−0.46 **	−0.47 **	−0.47 **	−0.16	−0.49 **	−0.48 **	/	/
Training (yes) * sexual minority (no)	−0.21	−0.21	−0.21	/	/	/	/	/
Adjusted R^2^	0.18	0.18	0.18	0.18	0.18	0.18	0.17	0.17

* *p* < 0.05, ** *p* < 0.01, *** *p* < 0.001; reference groups country = Sweden, profession = general practitioners. AFAB = Assigned female at birth.

## Data Availability

Restrictions apply to the availability of the data. Data was obtained from Transgender Europe and are available from the authors with the permission of Transgender Europe.

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
