# Peer review of "The Impact of Training in Transgender Care on Healthcare Providers Competence and Confidence: A Cross-Sectional Survey"

_healthcare, 2021, doi:10.3390/healthcare9080967_

Round 1
Reviewer 1 Report
General statement:
The study addresses the impact of trainings in providing care for members of the transgender community on HCP confidence and competence. The authors display a well-organized overview of current and past literature in the field and identify the gap. The methodology is appropriate for the research. Findings were clearly stated, and themes described. The limitation section provided a clear overview of sampling bias and recruitment strategies’ impact on the respondent pool. The discussion section provided some connection to current and past research in the area but would benefit from a deeper analysis.
It is important to keep in mind who the audience of this research study will be. Words are very powerful and therefore, have the potential to provide support for members of the transgender community or perpetuate existing bias in health care and further alienate this vulnerable population. I would like to commend the authors for taking on such an important topic and look forward to seeing it published after revisions.
Abstract:
The abstract includes all necessary sections.
Introduction:
The section was very organized, and the purpose of the study rooted in the existing literature regarding the training of HCP to work effectively with members of the transgender community. There are a few sections that require editing for implicit bias and language. Some word choices as highlighted in the attached document can actually perpetuate HCP attitude toward transgender clients.
It might be beneficial to provide some definition of the terms that are later used in the study (transgender vs transsexual) and edit for consistency (transgender client, transgender person, transgender people). The authors did not elaborate on the need for training to provide care for members of the transgender community. Additional research in that area would strengthen the purpose of the study. Authors need to be aware of the fact that the experiences of members of the LGB community differ greatly from those of the transgender community.
Methods:
The authors clearly outlined all procedural aspects in terms of coding and data analysis. The coding of participants' gender identity who did not list transgender as an option bears the potential of bias. It is strongly recommended that authors revisit this specific coding strategy.
Findings:
Statistical analysis appropriate. The tables demonstrated results clearly.
Discussion.
At the beginning of the discussion section, authors compared some of the results from the present study to findings from other research related to the area of study. However, this section could be strengthened by a more comprehensive analysis of results in regard to the reason behind the necessity of implementing trans-specific trainings. The discussion section would benefit from some editing for implicit bias as highlighted by the reviewer in the text document. The conclusion section is concise and clearly outlines the potential for future research and societal impact of the topic.
Please see attached PDF with comments

Reviewer 2 Report
See attachment

Reviewer 3 Report
- It is not clear if the authors followed the journal's style for citations since some references with three authors are showing up with only two authors cited. This may or may not be ok, depending on the journal style sheet.
- On page 5, 3.1 needs to be "Response" I think or "Respondents"
- In the following paragraph, the 52 trans HCPs were not included in the analysis on the assumption they would be better informed but it would seem a simple t-test might be able to check that assumption and inform readers if their hypothesis was correct. It probably will be but it would be so easy to go from "assumption" to "evidence" it's hard to understand why that possibility was ignored.
- Otherwise, a sound paper, if not terribly exciting (doctors need training and doctors want more training, with respect to transgender needs).
Round 2
Reviewer 2 Report
The manuscript has improved substantially.
However, I consider important the questions raised in the first review:
- Date of data collection: 2016-2017. Although it is a subject that the authors indicate that it has not evolved much, this is an assumption since, they have not verified if their data may differ in these years. Perhaps they can make a comparison with current data.
- Most importantly, the research methodology is a non-validated ad hoc questionnaire. I do not consider it acceptable to use a non-validated questionnaire in an impact publication.
- I consider that the qualitative methodology helps to acquire a deeper knowledge of the subject, since they have not used a validated questionnaire.
Author Response
Dear reviewer,
thank you for your feedback. We have adapted the manuscript accordingly. See our response in attach.
Kind regards,
Aisa Burgwal & Joz Motmans
